# Low-frequency neural activity reflects rule-based chunking during speech listening

**Peiqing Jin[1], Yuhan Lu[1], Nai Ding[1,2]\***

[1]Key Laboratory for Biomedical Engineering of Ministry of Education, College of Biomedical Engineering and Instrument Sciences, Zhejiang University, Hangzhou, China; [2]Research Center for Advanced Artificial Intelligence Theory, Zhejiang Lab, Hangzhou, China

**Abstract** Chunking is a key mechanism for sequence processing. Studies on speech sequences have suggested low-frequency cortical activity tracks spoken phrases, that is, chunks of words defined by tacit linguistic knowledge. Here, we investigate whether low-frequency cortical activity reflects a general mechanism for sequence chunking and can track chunks defined by temporarily learned artificial rules. The experiment records magnetoencephalographic (MEG) responses to a sequence of spoken words. To dissociate word properties from the chunk structures, two tasks separately require listeners to group pairs of semantically similar or semantically dissimilar words into chunks. In the MEG spectrum, a clear response is observed at the chunk rate. More importantly, the chunk-rate response is task-dependent. It is phase locked to chunk boundaries, instead of the semantic relatedness between words. The results strongly suggest that cortical activity can track chunks constructed based on task-related rules and potentially reflects a general mechanism for chunk-level representations.

**\*For correspondence:**
ding_nai@zju.edu.cn

**Competing interests:** The authors declare that no competing interests exist.

## Introduction

How the brain processes sequences is a central question in cognitive science and neuroscience, and speech is a classic example of complex and rapid sequences that the human brain can effectively process. In general, speech utterances such as sentences are not memorized sequences, but instead are proposed to reflect compositional processes which allow us to understand and produce countless new sentences never heard before (such as the one you are currently reading). Therefore, to derive the meaning of a speech sequence, the brain has to integrate information across words, the meaning of which are stored in long-term memory. Recent neurophysiological results have shown that when listening to speech, cortical activity is observed on multiple time scales that match the time scales of multiple levels of linguistic units, such as sentences, phrases, words, and syllables (***Brodbeck et al., 2018***; ***Broderick et al., 2018***; ***Ding et al., 2017***; ***Ding et al., 2018***; ***Keitel et al., 2018***; ***Makov et al., 2017***). Critically, cortical responses on the time scales of phrases and sentences are observed even when the phrasal and sentential boundaries are not cued by prosodic features or by the transitional probability between words.

The neural responses on the time scales of phrases and sentences have been taken as strong evidence that the brain applies grammatical rules to group words into chunks (***Ding et al., 2017***; ***Martin and Doumas, 2017***). Challenging this rule-based chunking interpretation, it has been argued that neural responses at the phrasal and sentential rates may not reflect neural construction of multiword chunks based on rules and can instead be explained by neural tracking of properties of individual words alone (***Frank and Yang, 2018***). In the English materials used in ***Ding et al. (2017)***, for example, all sentences have the same structure of adjective+noun+verb+noun, for example 'new

**eLife digest** From digital personal assistants like Siri and Alexa to customer service chatbots, computers are slowly learning to talk to us. But as anyone who has interacted with them will appreciate, the results are often imperfect.

Each time we speak or write, we use grammatical rules to combine words in a specific order. These rules enable us to produce new sentences that we have never seen or heard before, and to understand the sentences of others. But computer scientists adopt a different strategy when training computers to use language. Instead of grammar, they provide the computers with vast numbers of example sentences and phrases. The computers then use this input to calculate how likely for one word to follow another in a given context. "The sky is blue" is more common than "the sky is green", for example.

But is it possible that the human brain also uses this approach? When we listen to speech, the brain shows patterns of activity that correspond to units such as sentences. But previous research has been unable to tell whether the brain is using grammatical rules to recognise sentences, or whether it relies on a probability-based approach like a computer.

Using a simple artificial language, Jin et al. have now managed to tease apart these alternatives. Healthy volunteers listened to lists of words while lying inside a brain scanner. The volunteers had to group the words into pairs, otherwise known as chunks, by following various rules that simulated the grammatical rules present in natural languages. Crucially, the volunteers' brain activity tracked the chunks – which differed depending on which rule had been applied – rather than the individual words. This suggests that the brain processes speech using abstract rules instead of word probabilities.

While computers are now much better at processing language, they still perform worse than people. Understanding how the human brain solves this task could ultimately help to improve the performance of personal digital assistants.

---

plans gave hope'. Therefore, neural activity tracking lexical properties, such as part of speech information or lexical semantic information that distinguishes objects and actions, may appear to track sentences and phrases. For example, neural activity tracking verbs will occur at the sentence rate, since there is one verb per sentence. Consistent with this lexical property model, it has been shown that apparent neural tracking of sentences could be observed if the neural response independently represents each word using multidimensional features learned by statistical analysis of large corpora (*Frank and Yang, 2018*). In other words, neural populations that are tuned to lexical features of individual words may show activity that apparently tracks sentence structures.

Furthermore, it is well established that the neural response to a word depends on the context and the response amplitude is smaller if the word is semantically related to previous words (*Kutas and Federmeier, 2011*; *Lau et al., 2008*). In general, words within the same sentence are more related than words from neighboring sentences. Therefore, for a context-dependent neural response, its amplitude is expected to be stronger at the beginning of a sentence, leading to apparent neural tracking of sentences. This model considers semantic relatedness between consecutive words, but it does not consider the sentence structure: Semantic relatedness is evaluated the same way within and across sentence boundaries. Apparent sentence tracking behavior is generated since words within a sentence are more closely related.

The lexical property model and semantic relatedness model do not assume chunk-level representations and therefore provide different explanations for sentential/phrasal-rate responses than the rule-based chunking model. The rule-based chunking model, however, has additional flexibility, allowing the same sequence of words to be grouped differently based on different sets of rules. This flexibility is most clearly demonstrated when processing structurally ambiguous sequences. For example, 'sent her kids story books' can be chunked as 'sent [her kids] story books' or 'sent her [kids story books]' (*Shultz and Pilon, 1973*). For such structurally ambiguous sentences, the rule-based chunking model, but not the lexical property model or the semantic relatedness model, would predict different phrase-tracking responses when the sentence is chunked differently.

Notably, the three models introduced here are not exclusive, and neural encoding of lexical properties is a prerequisite to analyze the semantic relations between words or to build multi-word chunks. It is well established that the brain responses reflect encoding of lexical properties and semantic relations, but it remains debated whether the brain represents multi-word chunks. Therefore, the goal of the current study is to test whether neural activity on the time scales commensurate with multi-word chunks indeed indicates chunk-level neural representations or can be explained by the simpler word-level representations.

Here, we distinguished the rule-based chunking model from the word-based models by asking the listeners to chunk a word sequence based on different rules in two experimental conditions. The word sequence was a sequence of nouns that describe either living (L) things or nonliving (N) things (*Figure 1A*). The sequence had no syntactically defined chunks. Nevertheless, the participants explicitly learned artificial rules to chunk the sequence, and rules varied between two conditions. While participants performed the rule-based chunking task, their cortical responses were recorded using MEG. The word-level models, that is the lexical property model and the semantic relatedness model, predicted the same neural response when participants listened to the same word sequence

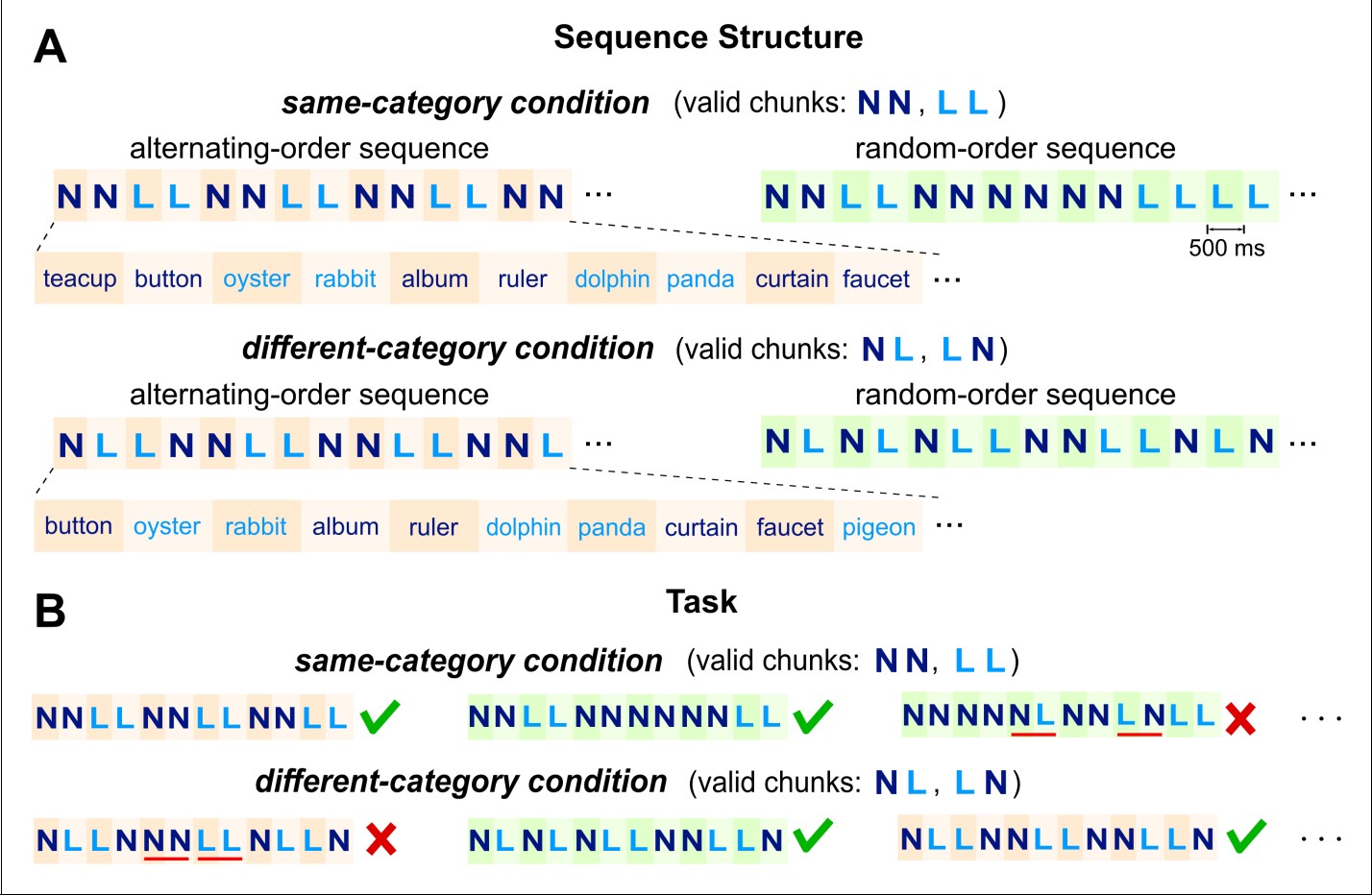

**Figure 1.** Stimuli and task. (**A**) Stimuli consist of isochronously presented nouns, which belong to two semantic categories, that is living (L) or nonliving (N) things. Two nouns construct a chunk and the chunks further construct sequences. In the same-category condition (upper panel), nouns in a chunk always belong to the same semantic category. In the different-category condition (lower panel), nouns in a chunk always belong to different categories. Sequences in each condition further divide into alternating-order (left) and random-order (right) sequences. The alternating-order sequence only differs by a time lag between the same- and different-category conditions. In the illustration, colors are used to distinguish words from different categories and the two words in a chunk but in the speech stimulus all words are synthesized in the same manner. (**B**) The task is to decompose each sequence into two-word chunks and detect invalid chunks. Three trials and the correct responses are shown for each condition (tick and cross for normal and outlier trials, respectively). Red underlines highlight the invalid chunks.

regardless of how the sequence was chunked. The rule-based chunking model, however, predicted chunk-dependent neural responses.

## Results

### Word sequences and model predictions

Participants were instructed to parse a sequence of words into chunks and each chunk consisted of two words. The words were drawn from two categories, that is living (L) and nonliving (N) things. The experiment contrasted two conditions in which the chunks were constructed based on different rules. In one condition, referred to as the same-category condition, the two words in a chunk belonged to the same semantic category (*Figure 1A*, upper panel). In the other condition, referred to as the different-category condition, the two words in a chunk were drawn from different categories (*Figure 1A*, lower panel). Based on these rules, there were two valid chunks in the same-category condition, that is LL and NN, and also two valid chunks in the different-category condition, that is NL and LN.

The same- and different-category conditions were presented in separate blocks. In each condition, participants were asked to parse the sequences into chunks, that is pairs of words, and were instructed about the rules that valid chunks followed. They had to judge if any invalid chunk was presented in a sequence (*Figure 1B*). The sequences in each condition ($N$ = 60) were further divided into two types, that is the alternating-order sequence ($N$ = 30) and the random-order sequence ($N$ = 30). In the alternating-order sequence, the two valid chunks in each condition were interleaved while in the random-order sequence the two valid chunks were presented in random order. Equal numbers of alternating-order sequences and random-order sequences were intermixed and presented randomly in each block. The behavioral correct rate was 85 ± 2% and 86 ± 2% for the same- and different-category conditions respectively (mean ± SEM across participants). The neural responses to these two types of sequences were separately analyzed. The alternating-order sequences allowed an intuitive comparison between the same- and different-category conditions, which would be detailed in the following. The random-order sequences were designed as fillers to avoid alternative strategies for the task (see Materials and methods for details), but model simulations in the following showed that they could also distinguish the three models.

Simulations of the neural responses to the alternating- and random-order sequences by the three models are shown in *Figure 2* for both the same- and different-category conditions. The lexical property model considers two neural populations that idealize tuning to living and nonliving word meanings, respectively. Since the two neural populations are anti-correlated, only the neural population tuned to living things is shown (*Figure 2AB*). The simulated neural response analyzed in the frequency domain demonstrates that neither the population shows a spectral peak at 1 Hz, that is the chunk rate, in the spectrum. In contrast to the lexical property model, the semantic relatedness model and the rule-based chunking model predict a 1 Hz response peak (red and green curves respectively in *Figure 2AB*).

A more fundamental difference between the semantic relatedness model and rule-based chunking model lies in their predictions about the 1 Hz response phase. The semantic relatedness model predicts a 180° phase difference between same- and different-category conditions, while the rule-based chunking model predicts a 0° phase difference between conditions (*Figure 2*). For the alternating-order sequence, these predictions are straightforward: These sequences are offset by one word between the same- and different-category conditions. Consequently, neural activity tracking semantic relatedness between words is offset by the duration of a word between conditions. For neural activity at 1 Hz, this time lag lead to a 180° phase difference. For the random-order sequence, although less straightforward, model simulation shows that the 1 Hz response has a 180° phase difference between conditions. For the rule-based model, however, the response is aligned with the chunk boundaries, which are not affected by the conditions and sequence types. Therefore, the rule-based model predicts the same response phase, that is a 0° phase difference, for the same- and different-category conditions.

In summary, the three models considered in this study lead to different predictions about the neural responses (*Figure 2AB*). Details about the model simulations are given in *Figure 2—figure supplement 1A*. The lexical property model and the semantic relatedness model assume ideal

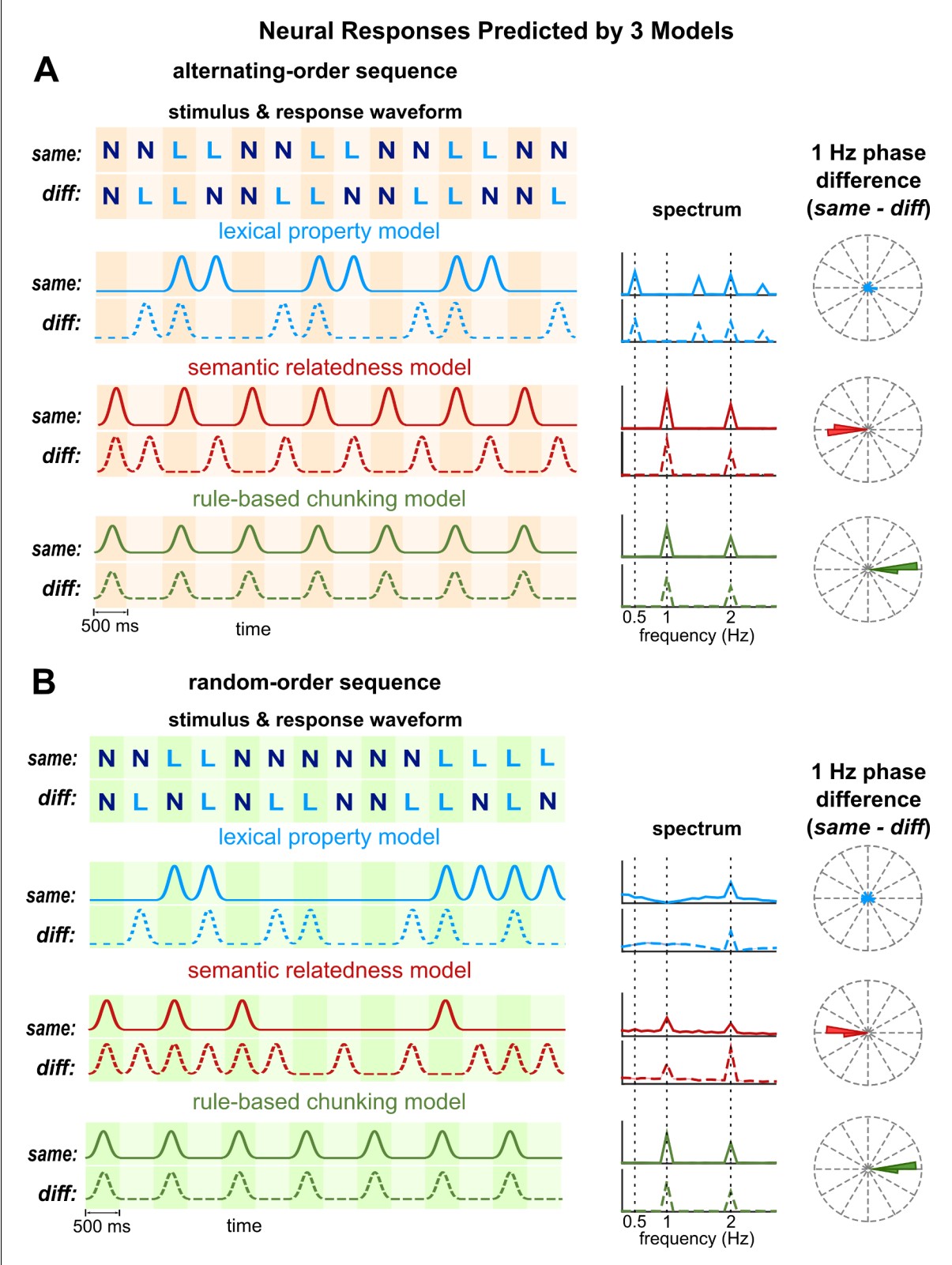

**Figure 2.** Model simulations. (**A**) Simulations for alternating-order sequences. The left panel illustrates the response waveform for example sequences. Neural responses predicted by the lexical property and semantic relatedness models differ by a time lag between same- and different-category conditions. The rule-based chunking model, however, predicts identical responses in both conditions. The middle panel shows the predicted spectrum averaged over all sequences in the experiment. The semantic relatedness model and the rule-based chunking model both predict a significant 1 Hz

*Figure 2 continued on next page*

*Figure 2 continued*

response, while the lexical property model predicts a significant response at 0.5 Hz and its odd order harmonics. The right panel shows predicted phase difference between same- and different-category conditions at 1 Hz. The phase difference predicted by the lexical property model is uniformly distributed. The semantic relatedness model predicts a 180° phase difference between conditions, while the rule-based chunking model predicts a 0° phase difference. (B) Simulations for random-order sequences. The semantic relatedness model and rule-based model predict a significant 1 Hz response. They generate different predictions, however, about the 1 Hz phase difference between conditions.

The online version of this article includes the following figure supplement(s) for figure 2:

**Figure supplement 1.** Model simulation.

tuning to living/nonliving things. In the following, we turn to the actual neural responses obtained using MEG and evaluate their consistency with the simulations made for the three different models.

## Rule-dependent neural tracking of Alternating-order sequences

The MEG responses were separately averaged for the same- and different-category conditions and the mean response was transformed to the frequency domain. We first analyzed the MEG responses to the alternating-sequences. The response spectrum averaged over all MEG gradiometers showed a clear peak at 1 Hz (*Figure 3A*, left two plots). The 1 Hz response power was significant in both conditions ($F_{32,64}$ = 5.8, p=$2.0 \times 10^{-9}$ and $F_{32,64}$ = 6.5, p=$2.1 \times 10^{-10}$ for the same- and different-category conditions respectively; F-test, FDR corrected). The 1 Hz spectral peak was consistent with the semantic relatedness model and the rule-based chunking model, but not with the lexical property model (*Figure 2A*). On top of the 1 Hz response peak, a 2 Hz response peak was clearly observed and was significant ($F_{32,64}$ = 45.8, p=$9.7 \times 10^{-33}$ and $F_{32,64}$ = 35.9, p=$1.4 \times 10^{-29}$ for the same- and different-category conditions respectively; F-test, FDR corrected). However, no significant peak was observed at 0.5 Hz ($F_{32,64}$ = 1.3, p=0.17 and $F_{32,64}$ = 1.0, p=0.52 for the same- and different-category conditions respectively; F-test, FDR corrected).

We then analyzed the phase difference between same- and different-category conditions at 1 Hz. In all the 306 MEG sensors, the phase difference averaged over participants was closer to 0° than 180° (*Figure 3E*), and in 258 sensors the effect was significant (p<0.01, bootstrap, see Materials and methods, FDR corrected). These results were consistent with the rule-based chunking model (*Figure 2A*). The phase difference averaged over all MEG sensors was significantly closer to 0° than 180° (p=$1 \times 10^{-4}$, bootstrap, see Materials and methods). The mean phase difference was 0.5° and the 99% confidence interval ranged from −22.9° to 18.7°. The main results on response spectrum and response phase difference could be reliably observed in individual participants (*Figure 3—figure supplement 1A*). To further illustrate the response phase difference in the time domain, we analyzed the response waveforms. The Principal Component Analysis (PCA) was employed to extract major response patterns from all MEG sensors. The first PC captured the MEG response to the sound onset (*Figure 3—figure supplement 1B*) and the second PC captured the 1 Hz response, which oscillate in phase in the same- and different-category conditions (*Figure 3F*).

## Rule-dependent neural tracking of Random-order sequences

The MEG responses to random-order sequences were analyzed the same way as the responses to alternating-order sequences, and the results were similar: In the response spectrum, a clear peak was observed at 1 Hz (*Figure 3A*, right two plots). The 1 Hz response power was significant in both conditions ($F_{32,64}$ = 4.5, p=$2.6 \times 10^{-7}$ and $F_{32,64}$ = 5.9, p=$1.5 \times 10^{-9}$ for the same- and different-category conditions respectively; F-test, FDR corrected). The 1 Hz response power was not significantly different between random- and alternating-order sequences ($F_{32,32}$ = 1.0, p=0.99 and $F_{32,32}$ = 1.1, p=0.90 for same- and different-category conditions respectively; F-test, FDR corrected; *Figure 3B*). The responses to random-order sequences also showed a significant 2 Hz response peak ($F_{32,64}$ = 48.5, p=$1.7 \times 10^{-33}$ and $F_{32,64}$ = 44.6, p=$2.0 \times 10^{-32}$ for the same- and different-category conditions respectively; F-test, FDR corrected). The peak at 0.5 Hz was not significant ($F_{32,64}$ = 0.75, p=0.81 and $F_{32,64}$ = 0.91, p=0.61 for the same- and different-category conditions respectively; F-test, FDR corrected). At 1 Hz, the phase difference between conditions was closer to 0° than 180° in all MEG sensors (*Figure 3E*), and the effect was significant in 196 sensors (p<0.01, bootstrap, see Materials and methods, FDR corrected). The phase difference averaged over all MEG sensors was

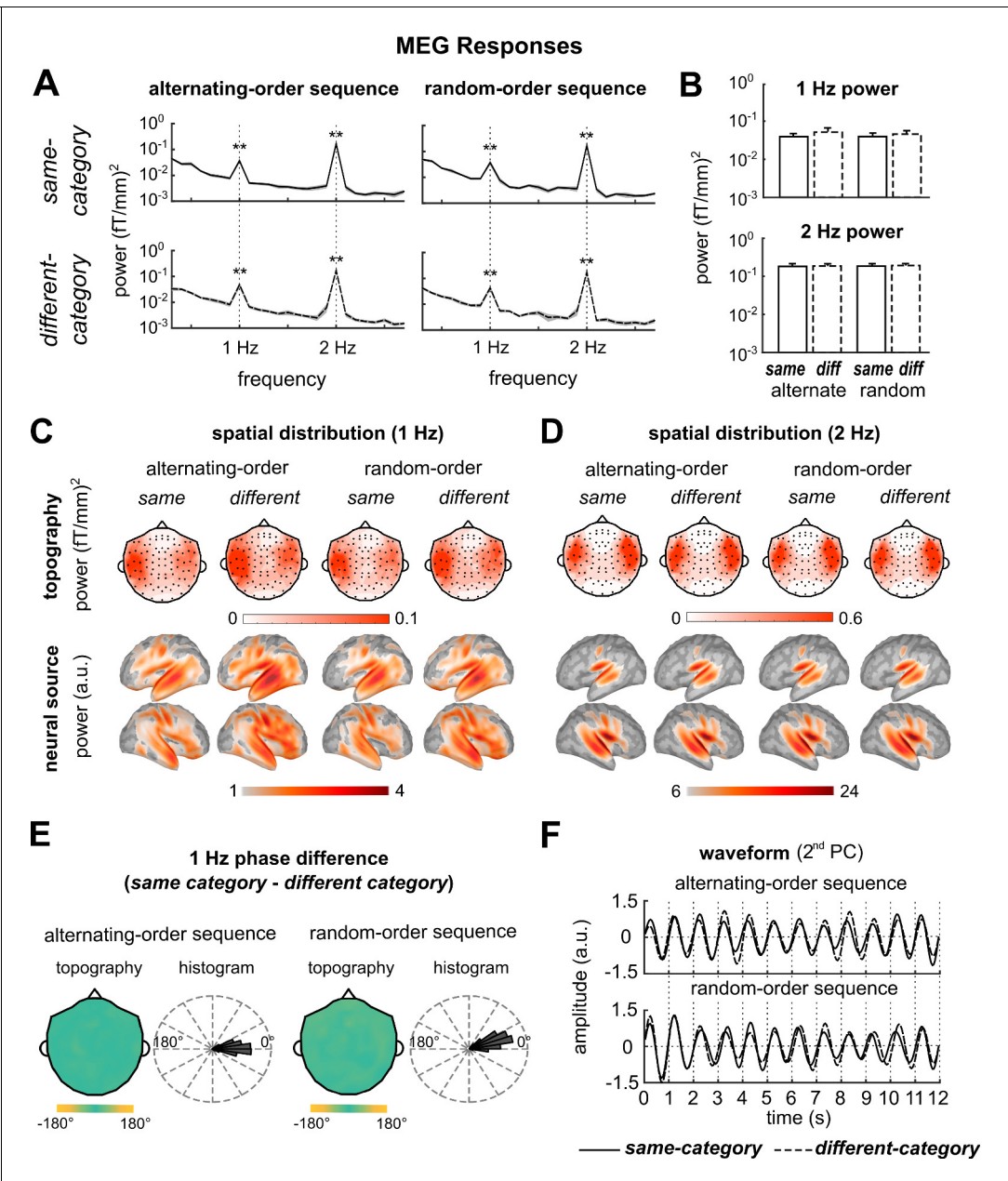

**Figure 3.** MEG responses. (**A**) The response power spectrum averaged over participants and MEG gradiometers. A 1 Hz response peak and a 2 Hz response peak are observed. The shaded area covered 1 SEM over participants on each side. (**B**) The 1 Hz and 2 Hz response power. There is no significant difference between conditions. (**C, D**) Response topography (gradiometers) and source localization results, averaged over participants. Sensors (shown by black dots) and vertices that have no significant response (p>0.05; F-test; FDR corrected) are not shown in the topography and localization results. The 1 Hz and 2 Hz responses both show bilateral activation. The neural source localization results are shown by the dSPM values and vertices with dSPM smaller than the min value in the color bar are not shown. (**E**) Phase difference between the same-category and different-category conditions at 1 Hz. The topography shows the distribution of phase difference across MEG sensors (one magnetometers and two gradiometers in the same position are circular averaged). The topography is circular averaged over participants. The histogram shows the phase difference distribution for all 306 MEG sensors. The phase difference is closer to 0° (predicted by the rule-based chunking model) than 180° (predicted by the semantic relatedness model). (**F**) Waveform averaged over trials and subjects (the second PC across MEG sensors). The waveform is filtered around 1 Hz and is highly consistent between the same- and the different-category conditions. *p<0.05, **p<0.005

The online version of this article includes the following figure supplement(s) for figure 3:

**Figure supplement 1.** Additional MEG results.

significantly closer to 0° than 180° (p=4 $\times$ 10$^{-4}$, bootstrap, see Materials and methods). The mean phase difference was 20.3° and the 99% confidence interval over participants ranged from −0.1° to 56.6°.

For both the neural responses to alternating-order and random-order sequences, in the response topography, the 1 Hz and 2 Hz responses showed bilateral activation (*Figure 3CD*). Neural source localization results confirmed that both the 1 Hz and 2 Hz responses were mainly generated from bilateral temporal and frontal lobes (*Figure 3CD*). An ROI analysis further revealed that the 1 Hz response peak was statistically significant in both temporal and frontal lobes ($F_{32,64} > 3.3$, p<1.9 $\times$ 10$^{-5}$ for all conditions; F-test, FDR corrected), and the response is stronger in the left superior temporal gyrus than the left inferior frontal gyrus ($F_{128,128} = 2.0$, p=7.1 $\times$ 10$^{-4}$; F-test, FDR corrected; *Figure 3—figure supplement 1C*).

## Discussion

How chunks are represented during sequence processing is a prominent question in psychology and cognitive neuroscience. Here, we demonstrate that low-frequency neural activity can track multi-word chunks that are mentally constructed based on artificial chunking rules. Critically, the artificial rules here can dissociate the properties of individual words from the chunk structures and therefore provide strong evidence that low-frequency neural activity can encode chunks.

Whether items in a sequence are mentally represented by hierarchically organized chunks is a heavily debated question in many cognitive domains, including language comprehension, action planning, and event perception (*Everaert et al., 2015*; *Frank et al., 2012*; *Lashley, 1951*). The multi-item chunks may be called schemas, scripts, or phrases in different fields. For example, for the routine behavior to prepare coffee, one view holds that it constitutes a schema with hierarchical structures: The schema of preparing coffee divides into sub-schemas such as adding sugar and adding milk, while each sub-schema further divides into finer-grained sub-schemas such as opening a package and pouring sugar into coffee (*Cooper and Shallice, 2006*). A contrasting view, however, is that hierarchical schemas are epiphenomenal and the internal cortical states generating basic actions are chained directly without any additional superordinate representation (*Botvinick and Plaut, 2004*). Similarly, for newly learned complex sequential processing tasks, some studies found that the underlying neural activity can be modeled by item-based state-transition models (*Mante et al., 2013*) while others provide evidence for chunk-level neural representations (*Geddes et al., 2018*; *Jiang et al., 2018*).

Chunk-based models and item-based models favor different kinds of neural implementations. Representations of chunks could be implemented by sustained neural activation throughout the duration of a chunk (*Fuster, 2001*). In other words, the neural representation of a chunk is synchronized to the onset and offset of a chunk. Consistent with this idea, both computational models (*Cooper and Shallice, 2006*; *Martin and Doumas, 2017*) and neural responses (*Ding et al., 2017*; *Nozaradan et al., 2011*) show activity matching the time scales of hypothetical chunks during routine behavior and language/auditory perception. For item-based models, cortical state is updated item by item, which does not explicitly predict neural synchronization to chunks. Indeed, in some cognitive domains only item-rate neural activity is observed. For example, when a zebra finch sings a multi-syllable song, no slow neural activity matching the time scales of multi-syllabic chunks is observed. Instead, bursts of transient neural responses occur at syllable onsets, which strongly support the item-based state transition model (*Long et al., 2010*). The zebra finch songs, however, are highly stereotyped. Future studies are needed to establish whether the chunk-rate responses observed here is limited to the processing of flexible complex sequences or limited to the primate brain.

A potential way to link chunk-based and item-based processing mechanisms is that they both exist in the brain but are implemented in different brain areas and applied to different tasks (*Goucha et al., 2017*). During language processing, for example, it has been proposed that item-based simple combination of words is implemented in the anterior temporal lobe (*Bemis and Pylkkanen, 2013*; *Brennan et al., 2012*) while rule-based chunk-level processing is implemented in the inferior frontal lobe (*Grodzinsky and Friederici, 2006*; *Grodzinsky and Santi, 2008*). Furthermore, experience may also affect the processing mechanism. For example, when participants have to perform a task based on rules that are learned explicitly, either rules in an executive control task or

grammar in an unknown language, it is barely controversial that rule-based processing occurs in the brain, since there is not enough exposure to build up a state transition model. Nevertheless, for routine behavior or native language processing, extensive exposure allows the learning of statistical relations and it becomes difficult to distinguish rule-based models and item-based state transition models.

In the current study, listeners explicitly apply artificial rules to group words into chunks, in contrast to previous studies in which the listeners can rely on implicit linguistic knowledge to group words into phrases. It remains elusive whether the brain relies on similar mechanisms for sequence chunking in different tasks. The primary motivation for a domain-general implementation of sequence-chunking tasks is that many sequence-chunking tasks share common computational principles. One such common computation is to find and encode the chunk boundaries. Consistent with a domain-general mechanism to encode chunk boundaries, studies have found similar EEG responses, that is the closure positive shift (CPS), at the prosodic phrasal boundaries in speech (*Li and Yang, 2009*; *Steinhauer et al., 1999*) and music phrasal boundaries (*Zhang et al., 2016*). Similarly, a transient increase in brain activity has been observed at event boundaries in movies (*Zacks et al., 2001*) and in non-speech sound sequences (*Chait et al., 2007*).

Another common computation in sequence-chunking tasks is to integrate information within a chunk. It has been suggested that low-frequency neural activity may also reflect sequential information integration within spoken phrases (*Ding et al., 2017*) and musical meters (*Nozaradan et al., 2011*). During sequential decision making, it has been more quantitatively demonstrated that neural activity reflects accumulation of sensory evidence (*Barascud et al., 2016*; *O'Connell et al., 2012*; *Shadlen and Shohamy, 2016*). The task in the current experiment also engages sequential decision making but, unlike most previous sequential decision tasks, it does not allow the decision variable to be updated incrementally. Here, the first word in a chunk, whether an L or N, does not contribute to the decision, and the decision variable should only be updated when the second word is compared with the first word. Since the decision variable is updated every other words, that is every chunk, it could potentially contribute to the chunk-rate response. Nevertheless, the bilateral topography of chunk-rate response is not compatible with the previous finding that decision-related responses concentrate in central MEG channels (*de Lange et al., 2010*). Furthermore, the decision-related centro-parietal positivity (CPP) in EEG is shown to be equivalent to the P300 (*O'Connell et al., 2012*; *Twomey et al., 2015*) while the MEG counterpart of the P300 does not show a bilateral topography either (*Mecklinger et al., 1998*). Therefore, the neural source of the chunk-rate response is not identical to the P300, the dominant decision signal in previous EEG/MEG studies, but it may still reflect sequential decision making since such responses are widely distributed as shown by intracranial neural recordings (*Gold and Shadlen, 2007*).

The idea that different sequence chunking tasks involve common computational principles does not necessarily imply that these principles are implemented in a common neural network. Nevertheless, there is indeed evidence for domain-general neural networks for sequence chunking. For example, functional MRI studies show that ventrolateral prefrontal cortex, including the Broca's area, is not only a core area for language processing but also activated by rule-based nonlinguistic sequential processing tasks (*Koechlin and Jubault, 2006*; *Novick et al., 2005*; *Thompson-Schill et al., 2005*). There are also studies, however, arguing for domain-specific sequence processing mechanisms, especially for the processing of language (*Fedorenko et al., 2011*). For example, it has also been shown that, for newly learned rules, rules of different complexity activate different parts of the frontal lobe, forming a posterior-to-anterior gradient (*Badre and Nee, 2018*; *Koechlin and Summerfield, 2007*). During language processing, however, syntactic rules of different complexity all activate Broca's area (*Jeon and Friederici, 2013*). These findings lead to the hypothesis that automatic processes and more controlled processes rely on distinct neural circuits (*Jeon and Friederici, 2015*). Based on this hypothesis, the chunk-level task in the current study and syntactic analysis may engage different parts of the frontal lobe. With the spatial resolution of MEG, we cannot precisely localize the neural source of the chunk-rate response. However, it is found that the frontal lobe activation is bilateral with no clear lateralization between hemispheres (*Figure 3—figure supplement 1C*), in contrast to the clearly left lateralized sentence-tracking response (*Ding et al., 2016*; *Sheng et al., 2019*). Similar to the sentence-tracking response, the chunk-rate response is also stronger in the temporal lobe than in the frontal lobe, suggesting that both sentences and chunks defined by temporary rules can drive large-scale chunk-rate responses in the temporal lobe.

The results in the current study show that low-frequency neural activity primarily tracks rule-defined chunks instead of semantic relatedness. The semantic relatedness hypothesis, however, is based on solid evidence in the literature. It builds on the priming effect in the psychological literature (*Tulving and Schacter, 1990*) and the neural adaptation effect in the neuroscience literature (*Grill-Spector et al., 2006*), and is related to the predictive coding hypothesis (*Bar, 2007*; *Friston, 2005*; *Tian and Poeppel, 2013*). It is well established that if a word is preceded by a semantically related word, it is processed faster (*Collins and Loftus, 1988*) and its neural response, especially the ERP N400 component and its MEG counterpart, is reduced (*Broderick et al., 2018*; *Kutas and Federmeier, 2011*; *Lau et al., 2009*). In this study, words from the same semantic category are more closely related than words drawn from different categories. Nevertheless, the categories used here are broad categories (e.g. animals or plants). In general, words from a broad category, for example animals, are only weakly related compared with words from a narrower category, for example birds (*Quinn and Kinoshita, 2008*; *Vigliocco et al., 2002*). A weak relationship between words predicts a weak priming effect on the neural response (*Federmeier and Kutas, 1999*), which may underlie why semantic relatedness between words does not drive a strong neural response.

Furthermore, previous studies have identified two kinds of semantic priming, that is automatic and strategic priming (*Neely, 1977*). Automatic priming can be caused by, for example semantic relatedness between words in long-term memory. Strategic priming, however, can actively predict upcoming words based on temporally learned association rules. Behavioral experiments have demonstrated a cross-category priming effect if the prime word from one category, for example tools, is known to predict target words from a different category, for example animals (*Neely, 1977*). In other words, participants can make use of association rules learned during an experiment to actively predict words that have no long-term semantic relationship with the prime word. Different from automatic priming that can occur with very short stimulus onset asynchrony (SOA) between words, strategic priming occurs when the SOA between the prime and target words is relatively long, e.g., >400 ms (*Hutchison, 2007*).

In the current study, the SOA between words is 500 ms, allowing strategic priming to occur. Furthermore, since the chunking rule remains the same in each block, listeners can prepare in advance about how to parse the sequences, making strategic predictions to occur more easily. Based on the knowledge about valid chunks, the semantic category of the second word in each chunk is fully predictable in both the same-category condition and the different-category condition. The first word in each chunk is also predictable in the alternating-order sequences but not predictable in the random-order sequences. Since the alternating-order sequences and the random-order sequences are mixed, predictability is generally lower for the first word than for the second word in each chunk. Therefore, for strategic predictions, the predictability of words correlates with the chunk structure. This kind of strategic predictions, however, is based on rule-based chunking instead of semantic relatedness stored in long-term memory.

Finally, the chunk-rate response in the current study is consistent with the rule-based chunking model. However, it may reflect the actual chunking process or downstream processes building on the multi-word chunks. After the chunk structure is parsed, the listener could synchronize their attention and predictions to the sequence. Previous studies have suggested that entrained neural oscillations may reflect both sequence parsing (*Ding et al., 2017*; *Kösem et al., 2016*; *Meyer and Gumbert, 2018*; *Meyer et al., 2016*; *Wang et al., 2017*) and temporal attention/prediction (*Jin et al., 2018*; *Morillon and Baillet, 2017*; *Rimmele et al., 2018*), and could causally modulate speech perception (*Kösem et al., 2018*; *Riecke et al., 2018*; *Zoefel et al., 2018*). The current results cannot distinguish which chunk-related process drives the chunk-rate response. What can, however, be concluded here is that the chunk-rate response cannot be fully accounted by neural tracking of individual words. Thus, the current study and previous studies (*Ding et al., 2017*) provide strong support to the notion that the brain can construct superordinate linguistic representations based on either long-term syntactic rules or temporary rules learned in an experiment.

## Materials and methods

### Participants

Sixteen participants took part in the study (19–27 years old; mean age 22.6; eight female). All participants were right-handed, with no self-reported hearing loss or neurological disorders. The experimental procedures were approved by the Research Ethics Committee of the College of Medicine, Zhejiang University (2019–047) and the Research Ethics Committee of Peking University (2019-02-05). The participants provided written consent and were paid.

### Words and sentences

All words were disyllabic words in mandarin Chinese and each syllable was a morpheme. For the noun sequences, each word was selected from a pool of 240 disyllabic concrete nouns. These concrete nouns equally divided into two categories, that is living (L) and nonliving (N) things. Living things further divided into 2 subcategories, that is animals (*N* = 60; e.g., monkey, panda) and plants (*N* = 60; e.g., tulip, strawberry). Nonliving things also divided into two subcategories, that is small manipulatable objects (*N* = 60; e.g., teacup, toothbrush) and large non-manipulatable objects (*N* = 60; e.g., playground, hotel). In each noun sequence, all living things were randomly drawn from a subcategory, that is animals or plants, and all nonliving things were also randomly drawn from a subcategory, that is manipulatable or non-manipulatable objects. Details about how the nouns constructed noun sequences are provided in the *Sequence Structure* section.

Each disyllabic word was independently synthesized by the iFLYTEK synthesizer (http://peiyin.xunfei.cn/; female voice, Xiaoying). All disyllabic words were adjusted to the same intensity and the same duration, that is 500 ms, following the procedure in *Ding et al. (2017)*. Within a word, no additional control was applied to the intensity and duration of individual syllables and coarticulation could exist between these syllables. Compared with speech materials in which each syllable was independently synthesized, the disyllabic words synthesized as a whole sounded more natural.

When constructing sequences, the synthesized disyllabic words were directly concatenated, without any additional pause in between. Therefore, words were isochronously presented at 2 Hz. For speech stimuli generated according to this procedure, each disyllabic word was an acoustically independent unit and larger chunks consisting of multiple words had no acoustically defined boundaries.

### Sequence structures

Pairs of nouns constructed chunks and chunks further constructed sequences. The experiment compared two conditions in which the chunks were constructed based on different rules. For the same-category condition, the two nouns in each chunk belonged to the same semantic category. For the different-category condition, however, the two nouns in each chunk were from different semantic categories. Since the study only considered two categories of words, there were two valid chunks in the same-category condition, that is LL and NN, and two valid chunks in the different-category, that is NL and LN. Each chunk was 1 s in duration.

Each sequence consisted of 12 chunks and therefore was 12 s in duration. In each sequence, the two valid chunks were concatenated in either an alternating order or a random order (*Figure 1A*). The alternating-order sequence in each condition had a fixed structure, repeating a four-words unit six times, that is NNLL for the same-category condition and NLLN for the different-category condition. The repeating four-words unit led to the 0.5 Hz rhythm in the lexical property model (*Figure 2*). In each random-order sequence, every chunk was randomly and independently chosen from the two valid chunks. After the category of each word was determined, the actual words were filled in. Each word was randomly drawn from a pool of 60 words (see *Speech Materials*), with an additional constraint that no word repeated in a sequence.

The alternating-order sequences had a highly regular structure, which led to a simple relationship between the alternating-order sequences in same- and different-category conditions: Any alternating-order sequence in the different-category condition could be converted to a same-category sequence by removing the first word in the sequence. Attributable to this property, the neural response phase could conveniently distinguish the word- and phrase-based models in *Figure 2*. Nevertheless, this property also gave rise to an alternative strategy that could detect invalid chunks based on the same set of rules in both the same- and different-category conditions. For this strategy,

the participants ignored the first word of each sequence in the different-category condition and treated the rest of the sequence as a same-category sequence. To eliminate this alternative strategy and to ensure that participants had to apply different rules in the same- and different-category conditions, the random-order sequences were designed as fillers to increase variability.

## Experimental procedures and tasks

Participants were familiarized with the synthesized words at the beginning of each experiment. In the familiarization session, after hearing a word, the participants pressed a key to see the word on a screen. Then, the participants could press one key to hear the word again or press another key to hear the next word.

In the MEG experiment, the same-category condition and the different-category condition were presented in separate blocks and the order of the two blocks was counterbalanced across participants. In each condition, 30 alternating-order sequences and 30 random-order sequences were mixed and presented in a random order. In eight alternating-order sequences and eight random-order sequences, a living noun in one chunk was switched with a nonliving noun in another chunk so that the two chunks were no longer valid (*Figure 1B*). These 16 sequences with invalid chunks were called outlier sequences. The outlier sequences ($N = 16$) and normal sequences ($N = 44$) were mixed and presented in a random order. The participants had a rest after listening to 30 sequences.

Before MEG recording, participants received training. The same-category condition was trained first, followed by the different-category condition. For each condition, participants were explicitly instructed that the sequence was constructed by bi-word chunks and explicitly instructed about how valid chunks were constructed based on living/nonliving things. They were told that any chunk that violated the chunk construction rule was invalid chunks that they had to detect. After receiving instructions, participants were familiarized with two normal sequences, followed by two outlier sequences. When listening to the outlier sequences, they were asked to verbally report the invalid chunks as soon as they heard them. The sequence could be replayed upon request. The participants then went through a practice session, which was the same as the MEG experiment, except that it was carried outside the MEG scanner.

During the practice session and during the MEG experiment, participants had to distinguish normal and outlier sequences and indicated their decisions by pressing different keys at the end of each sequence. After the key press, the next sequence was presented after a silent interval randomized between 1 s and 2 s (uniform distribution). The practice session ended after the participants made four correct responses in five consecutive sequences. The MEG experiment started after participants finished the practice session for the different-category condition.

## Data acquisition

Neuromagnetic responses were recorded using a 306-sensor whole-head MEG system (Elekta-Neuromag, Helsinki, Finland) at Peking University, sampled at 1 kHz. The system had 102 magnetometer and 204 planar gradiometers. Four MEG-compatible electrodes were used to record EOG at 1000 Hz. To remove ocular artifacts in MEG, the horizontal and vertical EOG were regressed out from the recordings using the least-squares method. Four head position indicator (HPI) coils were used to measure the head position inside MEG. The positions of three anatomical landmarks (nasion, left, and right pre-auricular points), the four HPI coils, and at least 200 points on the scalp were also digitized before experiment. For MEG source localization purposes, structural Magnetic Resonance Imaging (MRI) data were collected from all participants using a Siemens Magnetom Prisma 3 T MRI system (Siemens Medical Solutions, Erlangen, Germany) at Peking University. A 3-D magnetization-prepared rapid gradient echo T1-weighted sequence was used to obtain $1 \times 1 \times 1$ mm$^3$ resolution anatomical images.

## Data processing

In each condition, normal and outlier sequences were mixed and presented in a random order. However, only the neural responses to normal sequences were analyzed. Temporal Signal Space Separation (tSSS) was used to remove the external interference from MEG signals (*Taulu and Hari, 2009*). Since the current study only focused on responses at 0.5 Hz, 1 Hz, and 2 Hz, the MEG signals were bandpass filtered between 0.3 and 2.7 Hz using a linear-phase finite impulse response (FIR) filter (−6

dB attenuation at the cut-off frequencies, 10 s Hamming window). The frequency response curve of the FIR filter was compensated in the response spectrum.

The response during each sequence was extracted, downsampled to 20 Hz sampling rate, and was referred to as a trial.

The MEG signals were further denoised using a semi-blind source separation technique, the Denoising Source Separation (DSS). The DSS was a linear transform that decomposed multi-sensor MEG signals into components (*de Cheveigné and Parra, 2014*). The bias function of the DSS was chosen as the response averaged over trials within each condition. A common DSS for all conditions was derived based on the response covariance matrices averaged over conditions. The first six DSS components were retained and transformed back to the sensor space for further analysis. This DSS procedure was commonly used to extract cortical responses entrained to speech (*Ding et al., 2017*; *Zhang and Ding, 2017*).

To illustrate the response waveform, the PCA was employed to transform the 306-channel MEG data into components. Responses in the same- and different-category conditions and responses to the alternating- and random-order sequences were pooled in the PCA analysis. The first two PC were shown in *Figure 3—figure supplement 1B* and the 2nd PC, which captured the 1 Hz response, was filtered around 1 Hz and shown in *Figure 3F* (FIR filter with 2 s Hamming window, cut-off frequency: 0.75 and 1.25 Hz).

## Frequency-domain analysis

In the frequency-domain analysis, to avoid the response to the sound onset, the response during the first two seconds of each trial were removed. Consequently, the neural response was 10 s in duration for each trial. The average of all trials was transformed into the frequency domain using the Discrete Fourier Transform (DFT) without any additional smoothing window. The frequency resolution of the DFT analysis was 1/10 Hz. If the complex-valued DFT coefficient at frequency $f$ was denoted as $X(f)$, the response power and phase were $|X(f)|^2$ and $\angle X(f)$, respectively. The DFT was separately applied to each MEG sensor. For the MEG response power analysis, responses from the two collocated gradiometers were always averaged. When showing the spectrum, all MEG gradiometers were averaged. For the phase analysis, all magnetometers and gradiometers were separately analyzed. The circular mean was used to average the neural response phase over participants or sensors.

## Source localization

The MEG responses averaged over trials were mapped into source space using cortex constrained minimum norm estimate (MNE) (*Hämäläinen and Ilmoniemi, 1994*), implemented in the Brainstorm software (*Tadel et al., 2011*). The T1-weighted MRI images were used to extract the brain volume, cortex surface, and innermost skull surface using the Freesurfer software (http://surfer.nmr.mgh.harvard.edu/). In the MRI images, the three anatomical landmarks (nasion, left, and right pre-auricular points) were marked manually. Both three anatomical landmarks and digitized head points were used to align the MRI images with MEG sensor array. The forward MEG model was derived based on the overlapping sphere model (*Huang et al., 1999*). The identity matrix was used as noise covariance. Source-space activation was measured by the dynamic statistical parametric map (dSPM) (*Dale et al., 2000*) and the value was in arbitrary unit (a.u.). Individual source-space responses, consisting of 15,002 elementary dipoles over the cortex, was rescaled to the ICBM 152 brain template (*Fonov et al., 2011*) for further analyses.

## Source-space region of interest (ROI) analysis

Two ROIs were defined in each hemisphere. A frontal-lobe ROI included the pars opercularis and pars triangularis, and a temporal-lobe ROI included the superior temporal area. Anatomical areas are defined according to an automated landmark-based registration algorithm (*Desikan et al., 2006*). In source space, the response of all dipoles were transformed to the frequency domain, and the 90th percentile of response power was calculated for each ROI, at each frequency. When comparing the response between ROIs, the results were averaged over the alternating-order and random-order sequences and over the same-category and different-category conditions.

## Model simulations

Pulse sequence: In all three models, the smallest unit being considered was the word, and the model output was updated word by word. Therefore, in the simulations, each model was first simulated using a pulse sequence (*Figure 2—figure supplement 1A*), in which a pulse was placed at the onset of each word and the pulse amplitude was described in the following. The lexical property model and the semantic relatedness model were simulated based on lexical features. For the model illustrated in *Figure 2*, only two features were considered, that is living and nonliving things. Each feature took a binary value, that is one when the feature was present and 0 otherwise, and the pulse amplitude for each word equaled this binary value. The semantic relatedness model built on the lexical property model: The semantic relatedness between the current word and the previous word was characterized by the correlation coefficient between lexical representations (*Broderick et al., 2018*). The correlation coefficient was a scalar. Additionally, since the neural response to a stimulus is usually weaker instead of stronger if the stimulus is preceded by a similar stimulus, we used one minus the correlation coefficient to modulate a pulse sequence. In the rule-based chunking model, pulses of unit amplitude were placed at the chunk onset.

## Simulate neural response waveform

The neural responses were smooth waveforms rather than sharp pulses. Therefore, neural response waveforms were further simulated by convolving the pulse sequence with a response function, which was a 500 ms duration Gaussian window. Here, the rule-based chunking model was simulated by a response time locked to the chunk onset. In general, however, the model only assumed a consistent response within the duration of a chunk. Results in *Figure 2—figure supplement 1B* confirmed that the key predictions of the model were not affected by the waveforms.

## Statistical tests

### Spectral peak

An F-test was used to test if the power at a target frequency $f_T$ was significantly higher than the power in neighboring frequency bins (one bin on each side). The power ratio was defined as

$$PR(f_T) = 2 \sum_{1 \leq k \leq N} |X_k(f_T)|^2 / \sum_{1 \leq k \leq N} (|X_k(f_T - \Delta f)^2 + |X_k(f_T + \Delta f)|^2),$$

where $X$ was the complex-valued DFT coefficient defined in *Frequency-domain analysis* section, $f_T \pm \Delta f$ denoted the two neighboring frequency bins, and $k$ denoted data from the $k^{th}$ participant.

Under the null hypothesis, that is no difference between the power at the target and neighboring frequency bins, the power ratio was subject to an F(2$N$,4$N$) distribution for a single-sensor recording averaged over $N$ independent participants (*Dobie and Wilson, 1996*). Here, response power was calculated over sensors or dipoles and the degree of freedom would further increase if the sensors were not fully correlated. However, since it was difficult to quantify the increase in degree of freedom, we conservatively assumed that the power ratio remained following an F(2$N$,4$N$) distribution. The significance test was applied to the response power at the 0.5 Hz, 1 Hz and 2 Hz. A false discovery rate (FDR) correction was applied to these frequencies.

### Power difference between conditions

The F-test was used to compare the power at a target frequency between conditions. The response power comparison was performed between two conditions in which the response power was both significant at target frequency. The power ratio between conditions was subject to an F(2$N$,2$N$) distribution, where $N$ was the number of participants. Only in the ROI analysis, since the responses were averaged over the alternating-order and random-order sequences, and over the same-category and different-category conditions, the power ratio test was based on an F(8$N$,8$N$) distribution.

### Response phase

A test based on bias-corrected and accelerated bootstrap (*Efron and Tibshirani, 1994*) was used to test whether the response phase difference was closer to 0° or 180°. In the bootstrap procedure, all the participants were resampled with replacement 100,000 times. The test was two-sided. If the resampled phase difference was closer to 0° for $A$ times, the significance level was 2 min($A$ + 1,

$100001 - A)/100001$. Furthermore, the 99% confidence interval of the phase difference was also calculated based on the resampled data. It was measured by the smallest angle that could cover 99% of the resampled phase difference.

## Post-hoc effect size calculation

On top of the showing individual results in *Figure 3—figure supplement 1A*, an effect size analysis was applied to the 1 Hz spectral peak to validate that the sample size was appropriate. In this analysis, we applied a paired t-test to compare the power at the target frequency and the power averaged over two neighboring frequencies (both in dB scales). Such a t-test had weaker power than the F-test (*Dobie and Wilson, 1996*) but was easy to calculate the effect size using the G*Power Version 3.1 (*Faul et al., 2007*). We calculated d and Power based on the mean and standard deviation (reported in *Supplementary file 1*). For the effect size observed in the data set, the study was powerful with the described sample population and the $\alpha$ level of 0.05.

# Acknowledgements

We thank Cheng Cheng for excellent assistant in data collection, Jiajie Zou for discussion and generation some of the speech materials, Stefan L Frank, Lucia Melloni, Lang Qin, and David Poeppel for thoughtful comments on previous versions of the manuscript. Work supported by National Natural Science Foundation of China 31771248 (ND), Major Scientific Research Project of Zhejiang Lab 2019KB0AC02 (ND), Zhejiang Provincial Natural Science Foundation of China LY20C090008 (PJ), and Fundamental Research Funds for the Central Universities (ND).

# Additional information

### Funding

| Funder | Grant reference number | Author |
| --- | --- | --- |
| National Natural Science Foundation of China | 31771248 | Nai Ding |
| Major Scientific Research Project of Zhejiang Lab | 2019KB0AC02 | Nai Ding |
| Fundamental Research Funds for the Central Universities | | Nai Ding |
| Zhejiang Provincial Natural Science Foundation of China | LY20C090008 | Peiqing Jin |

The funders had no role in study design, data collection and interpretation, or the decision to submit the work for publication.

### Author contributions

Peiqing Jin, Data curation, Software, Formal analysis, Validation, Investigation, Visualization, Methodology, Writing - original draft; Yuhan Lu, Software, Formal analysis, Validation, Visualization; Nai Ding, Conceptualization, Resources, Funding acquisition, Validation, Visualization, Methodology, Writing - original draft, Project administration, Writing - review and editing

### Author ORCIDs

Peiqing Jin https://orcid.org/0000-0002-5302-4079
Yuhan Lu https://orcid.org/0000-0003-2684-1484
Nai Ding https://orcid.org/0000-0003-3428-2723

### Ethics

Human subjects: The experimental procedures were approved by the Research Ethics Committee of the College of Medicine, Zhejiang University (2019-047) and the Research Ethics Committee of Peking University (2019-02-05). The participants provided written consent and were paid.

Decision letter and Author response
Decision letter https://doi.org/10.7554/eLife.55613.sa1
Author response https://doi.org/10.7554/eLife.55613.sa2

## Additional files

### Supplementary files

• Source data 1. Preprocessed MEG data and analysis code.

• Supplementary file 1. 1 Hz power effect size.

• Transparent reporting form

### Data availability

The MEG data and analysis code (in MatLab) are available in Source data 1.

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
