## [Decision Letter]

**Acceptance summary:**

We believe that your work makes an important and timely contribution to the current debate regarding cortical responses to word-level features. With a smart experimental design, you clearly show that the mental chunking of word sequences contributes to shaping the neural responses.

**Decision letter after peer review:**

Thank you for submitting your article "Low-frequency neural activity reflects rule-based chunking during speech listening" for consideration by *eLife*. Your article has been reviewed by three peer reviewers, including Tobias Reichenbach as the Reviewing Editor and Reviewer #1, and the evaluation has been overseen by Barbara Shinn-Cunningham as the Senior Editor.

The reviewers have discussed the reviews with one another and the Reviewing Editor has drafted this decision to help you prepare a revised submission.

Summary:

The manuscript describes an experiment that assesses the importance of mental “chunking” of groups of words during speech listening for neural responses. This is a very timely investigation, since there is a current debate regarding cortical responses to word-level features, such as syntactic and semantic ones. A difficulty with the observations has been that the data could be explained by different models, such as models based on syntactic features or on word similarity. The authors use a clever design in which participants are asked to chunk the same stimuli in two different ways. They show that the neural responses change depending on how the chunking is done, although the underlying stimulus remains essentially the same.

Essential revisions:

1) The authors employ both regularly alternating and irregularly alternating sequences, but the case for considering these two types is not strongly made. The former are said to be "intuitive", and the second useful as "fillers" which is not strongly convincing. Is there something more to be gained (e.g. as suggested in the final paragraph of the Discussion)? Including both makes the story more complex, and the figures busier, so it is worth making the most of it.

2) The main results rely on the phase difference of the response at 1 Hz between the two categories “same” and “different”. But the significance of these phase differences is never assessed. They certainly seem to differ strongly from a uniform phase distribution, but this should nonetheless be confirmed through appropriate statistical testing. Moreover, the phase difference in the random-order sequence does seem to be centered slightly away from 0. Is that indeed the case, and if so, how could this slight shift away from zero be explained? Moreover, could one show time-domain plots of the trial-averaged response, in addition to, the frequency-domain analysis, to clearly make the point regarding the different phases?

3) The chunking, and the associated neural response, is not uniquely tied to speech processing. Instead, the task of the chunking of the word groups is an artificial one that could likely be applied to other auditory stimuli, such as groups of different notes. In particular, more general decision-making signals might play a role in the observed neural response. The subjects have to concentrate in this task and make a covert decision on each pair of words, potentially leading to a neural response that lines up with the chunks. Such a neural response could resemble a P3b (see, e.g., work from O'Connell and Kelly on the centro-parietal positivity decision-making signal and its relationship to the P3b component). Please discuss these issues in greater detail.

4) Related to the above, the authors discuss previous literature on chunking and its relationship to the frontal lobes. Why does the data presented here not show a contribution from the frontal lobes? It would also be good to compare the neural responses described in the manuscript to those reported in the author's previous publication (Ding et al., 2016b).

5) The Discussion ends relatively abruptly. Please add a paragraph that gives a summary and/or outlook.

[Editors' note: further revisions were suggested prior to acceptance, as described below.]

Thank you for resubmitting your work entitled "Low-frequency neural activity reflects rule-based chunking during speech listening" for further consideration by *eLife*. Your revised article has been evaluated by Barbara Shinn-Cunningham (Senior Editor), a Reviewing Editor, and the original reviewers.

The manuscript has been improved but there are some remaining issues that need to be addressed before acceptance, as outlined below:

Essential revisions:

Regarding the analysis of the statistical significance of the phase differences at 1 Hz, the authors now write that the phase differences were closer to 0 than to 180 degrees in all MEG sensors, and significantly so in 205 sensors. But for the test for statistical significance, they write that they have not corrected for multiple comparisons. Given the large number of MEG sensors, any result without correction for multiple comparisons appears meaningless. The authors should carry out an appropriate correction, and only report the results after this correction.

---

## [Author Response]

Essential revisions:1) The authors employ both regularly alternating and irregularly alternating sequences, but the case for considering these two types is not strongly made. The former are said to be "intuitive", and the second useful as "fillers" which is not strongly convincing. Is there something more to be gained (e.g. as suggested in the final paragraph of the Discussion)? Including both makes the story more complex, and the figures busier, so it is worth making the most of it.

The alternating-order sequences facilitate the comparison between same- and different-category conditions, since the alternating-order sequences in the 2 conditions are only offset by 1 word. Nevertheless, the highly regular structure of the alternating-order sequences allows the participants to detect invalid chunks using multiple strategies. Therefore, random-order sequences are introduced to avoid alternative strategies and force the participants to detect invalid chunks using different rules in the same- and different-category conditions.

We apologize that this critical information was not clearly conveyed in the previous manuscript. We have now added the following to Results:

“The random-order sequences were designed as fillers to avoid alternative strategies for the task (see Materials and methods for details).”

and the following to Materials and methods:

“The alternating-order sequences had a highly regular structure, which led to a simple relationship between the alternating-order sequences in same- and different-category conditions: Any alternating-order sequence in the different-category condition could be converted to a same-category sequence by removing the first word in the sequence. Attributable to this property, the neural response phase could conveniently distinguish the word- and chunk-based models in Figure 2. Nevertheless, this property also gave rise to an alternative strategy that could detect invalid chunks based on the same set of rules in both the same- and different-category conditions. For this strategy, the participants ignored the first word of each sequence in the different-category condition and treated the rest of the sequence as a same-category sequence. To eliminate this alternative strategy and to ensure that participants had to apply different rules in the same- and different-category conditions, the random-order sequences were designed as fillers to increase variability.”

2) The main results rely on the phase difference of the response at 1 Hz between the two categories “same” and “different”. But the significance of these phase differences is never assessed. They certainly seem to differ strongly from a uniform phase distribution, but this should nonetheless be confirmed through appropriate statistical testing.

We have now added a significance test on the phase difference and also reported the 99% confidence interval of the phase difference.

“In all the 306 MEG sensors, the phase difference averaged over participants was closer to 0° than 180° (Figure 3E), and in 261 sensors the effect was significant (P < 0.01, bootstrap, see Materials and methods, not corrected for multiple comparisons). These results were consistent with the rule-based chunking model (Figure 2A). The phase difference averaged over all MEG sensors was significantly closer to 0° than 180° (P = 1×10^-4^, bootstrap, see Materials and methods). The mean phase difference was 0.5° and the 99% confidence interval ranged from -22.9° to 18.7°.”

“At 1-Hz, the phase difference between conditions was closer to 0° than 180° in all MEG sensors (Figure 3E), and the effect was significant in 205 sensors (P < 0.01, bootstrap, see Materials and methods, not corrected for multiple comparisons). The phase difference averaged over all MEG sensors was significantly closer to 0° than 180° (P = 4×10^-4^, bootstrap, see Materials and methods). The mean phase difference was 20.3° and the 99% confidence interval over participants ranged from -0.1° to 56.6°.”

Moreover, the phase difference in the random-order sequence does seem to be centered slightly away from 0. Is that indeed the case, and if so, how could this slight shift away from zero be explained?

The phase difference for the random-order sequences slightly deviated from 0° but it was certainly closer to 0° than 180°. We have now reported the mean phase difference and its 99% confidence interval in Results, so that the readers could clearly see that the mean phase difference deviates from 0° but the 99% confidence interval still includes 0°. Furthermore, the new time course analysis revealed no clear phase lag between conditions either (Figure 3F). Therefore, we did not discuss this phase deviation since it was not a strong or reliable effect.

Moreover, could one show time-domain plots of the trial-averaged response, in addition to, the frequency-domain analysis, to clearly make the point regarding the different phases?

We have added time-domain results in Figure 3F and Figure 3—figure supplement 1C. We extracted the first 2 Principal Components (PC) based on the 306 MEG sensors. The 1^st^ PC captured the response to sound onset and the 2-Hz response (Figure 3—figure supplement 1C), while the 2^nd^ PC captured the 1-Hz response of interest. The 2^nd^ PC, when filtered around 1 Hz, clearly showed that the responses in same- and different-category conditions oscillated in phase (Figure 3F).

3) The chunking, and the associated neural response, is not uniquely tied to speech processing. Instead, the task of the chunking of the word groups is an artificial one that could likely be applied to other auditory stimuli, such as groups of different notes. In particular, more general decision-making signals might play a role in the observed neural response. The subjects have to concentrate in this task and make a covert decision on each pair of words, potentially leading to a neural response that lines up with the chunks. Such a neural response could resemble a P3b (see, e.g., work from O'Connell and Kelly on the centro-parietal positivity decision-making signal and its relationship to the P3b component). Please discuss these issues in greater detail.

Indeed, the chunk detection task is a sequential detection task and we have added discussions on how the chunk response may potentially relate to decision-making signals in the brain.

“During sequential decision making, it has been more quantitatively demonstrated that neural activity reflects accumulation of sensory evidence (Barascud et al., 2016; O'Connell et al., 2012; Shadlen and Shohamy, 2016). […] Therefore, the neural source of the chunk-rate response is not identical to the P300, the dominant decision signal in previous EEG/MEG studies, but it may still reflect sequential decision making since such responses are highly distributive as shown by intracranial neural recordings (Gold and Shalden, 2007).”

4) Related to the above, the authors discuss previous literature on chunking and its relationship to the frontal lobes. Why does the data presented here not show a contribution from the frontal lobes? It would also be good to compare the neural responses described in the manuscript to those reported in the author's previous publication (Ding et al., 2016b).

Thanks for raising this point. There is actually weak activation in the frontal lobe. However, similar to the response to sentences, the chunk-rate response is stronger in the temporal lobe. We have now added a ROI analysis to separately show the activation in temporal and frontal ROIs (Figure 3—figure supplement 1C) and added the following in to Results and Discussion.

“An ROI analysis further revealed that the 1-Hz response peak was statistically significant in both temporal and frontal lobes (F32,64 > 3.3, P < 1.9×10-5 for all conditions; F-test, FDR corrected), and the response is stronger in the left superior temporal gyrus than the left inferior frontal gyrus (F128,128 = 2.0, P = 7.1×10-4 ; F-test, FDR corrected; Figure 3—figure supplement 1C).”

“With the spatial resolution of MEG, we cannot precisely localize the neural source of the chunk-rate response. However, it is found that the frontal lobe activation is bilateral with no clear lateralization between hemispheres (Figure 3—figure supplement 1C), in contrast to the clearly left lateralized sentence-tracking response (Ding et al., 2016b; Sheng et al., 2019). Similar to the sentence-tracking response, the chunk-rate response is also stronger in the temporal lobe than in the frontal lobe, suggesting that both sentences and chunks defined by temporary rules can drive large-scale chunk-rate responses in the temporal lobe.”

5) The Discussion ends relatively abruptly. Please add a paragraph that gives a summary and/or outlook.

Thanks for the suggestion and we have now reorganized the Discussion section, which now ends with the following paragraph.

“Finally, the chunk-rate response in the current study is consistent with the rule-based chunking model. […] Thus, the current study and previous studies (Ding et al., 2016a) provide strong support to the notion that the brain can construct superordinate linguistic representations based on either long-term syntactic rules or temporary rules learned in an experiment.”

[Editors' note: further revisions were suggested prior to acceptance, as described below.]

Essential revisions:Regarding the analysis of the statistical significance of the phase differences at 1 Hz, the authors now write that the phase differences were closer to 0 than to 180 degrees in all MEG sensors, and significantly so in 205 sensors. But for the test for statistical significance, they write that they have not corrected for multiple comparisons. Given the large number of MEG sensors, any result without correction for multiple comparisons appears meaningless. The authors should carry out an appropriate correction, and only report the results after this correction.

We have now used the false discovery rate (FDR) correction for multiple comparisons.

“In all the 306 MEG sensors, the phase difference averaged over participants was closer to 0° than 180° (Figure 3E), and in 258 sensors the effect was significant (P < 0.01, bootstrap, see Materials and methods, FDR corrected).”

“At 1-Hz, the phase difference between conditions was closer to 0° than 180° in all MEG sensors (Figure 3E), and the effect was significant in 196 sensors (P < 0.01, bootstrap, see Materials and methods, FDR corrected).”